# RotPruner: Large Language Model Pruning in Rotated Space

## Abstract

Network pruning is a crucial technique for compressing large language models with billions of parameters, aiming to reduce memory and computational costs with minimal performance degradation. However, existing pruning methods for LLMs often focus on heuristic metrics or layer-wise reconstruction losses, neglecting the impact on the overall model output, which can lead to suboptimal result. Additionally, these methods operate directly on the original weight and activation spaces, which may not be ideal for pruning. In this paper, we propose that the original parameter space is not optimal for pruning and present a novel training-based pruning framework called RotPruner. RotPruner rotates the spaces of weight matrices and activations in linear layers, and applies existing pruning methods in a rotated space that is more suitable for pruning. We introduce an efficient algorithm to identify an appropriate rotation that preserves the performance of pruned LLMs. RotPruner is capable of integrating with other pruning methods and supporting unstructured, semi-structured, and structured pruning. We evaluate RotPruner on several large language models, including OPT, LLaMA-2, and LLaMA-3, and demonstrate state-of-the-art performance on both language modeling and zero-shot tasks.

## 1 Introduction

Recently, Large Language models (LLMs) have became a milestone in natural language processing, achieving great results in various tasks (Zhao et al., 2023). However, the success of these models results from an increase in scale and computational complexity, making the storage and time consuming of LLMs challenging. Model compression, as a post-training technique, has arouse great interest since it can reduce the memory and computational requirements of these models.

Model compression techniques usually include three types: distillation, pruning and quantization (Zhu et al., 2023; Gholami et al., 2022; Hoefler et al., 2021). In this work, we focus on pruning, which sets the several elements in the weight matrices of model to zero. Traditional pruning techniques often requires a post-pruning re-training to recover the performance after pruning (Ma et al., 2023; Huang et al., 2020; Han et al., 2015). However, this is challenging in LLMs due to its model size. To address this limitation, post-training method without re-retaining, such as Wanda (Sun et al., 2023) and SparseGPT (Frantar & Alistarh, 2023) are proposed.

Current pruning methods face two major challenges. First, traditional pruning methods focus on heuristic metric or individual layer's reconstruction loss and ignore inter-layer interaction, leading a high accumulative error. In contrast, block-wise, or model-wise pruning, considering a block's or the whole model's reconstruction loss, can reduce the error accumulation. The larger the pruning group, the more difficult the optimization. Second, current pruning methods run the algorithm in the original weight space, which is not the optimal space to prune. Changing the pruning space will give a better result of the same pruning method.

To address the above challenges, we introduce RotPruner, a novel training-based pruning framework of LLMs. Figure 1 provides the overview of our method, which applies rotation matrix to activations and weights, and runs pruning method on the rotated activations and weights. This approach does not update the weights and thus can preserve the knowledge of the pretrained model. On the rotated weights and activation, our method can leverage any other one-shot pruning methods and can realize both structured and unstructured pruning. Our further exploration finds that if we run the pruning

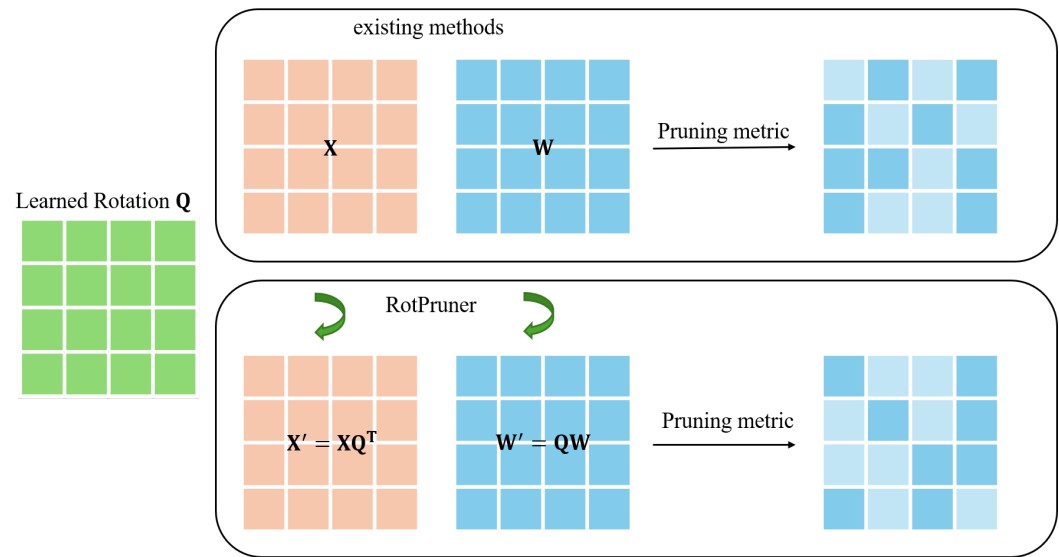

Figure 1: Overview of RotPruner. RotPruner first applies rotation to activation $X$ and weight $W$, and conduct pruning method on the rotated space. The rotation matrix is learned via cayley SGD.

method on random rotated space, the performance is catastrophic. Therefore, we propose to learn the rotation matrices utilizing cayley SGD (Li et al., 2020), which is an efficient optimizer for training orthonormal matrices, to optimize the rotation matrices to minimize the loss of the pruned network. The weights of the model are fixed and thus this do not change the result of dense model.

We empirically evaluate our method on widely adopted open-source LLM: OPT (Zhang et al., 2022), LLaMA-2 (Touvron et al., 2023) and LLaMA-3 (Dubey et al., 2024) families in the setting of un-structured, 2:4 semi-structured and structured pruning. Our approach exceeds the performance of state-of-the-art methods such as SparseGPT (Frantar & Alistarh, 2023), Wanda (Sun et al., 2023) and SliceGPT (Ashkboos et al., 2024a) across various language benchmarks. We show that the space of the original weight matrices is not the optimal one to be pruned on and introduce an approach to find a better pruning space.

## 2 RELATED WORK

### 2.1 NETWORK PRUNING

Network pruning is a widely used technique to reduce the model size and speed up the computation of neural network by generating sparse weight matrices. Pruning can be categorized into unstruc-tured, semi-structured, and structured pruning based on different sparsity patterns. Unstructured pruning eliminates the entries in the weight matrix without any structured pattern. While it can de-crease the entries of network, it can not get any inference speedup. Semi-structured pruning with N:M sparsity (Zhou et al., 2021) requires N non-zero entries in every continuous M weights. It can leverage NVIDIA's Sparse Tensor Cores to accelerate matrix computation. Structured pruning elim-inates the entries in entire rows or columns in the weight matrix and can reduce the dimension of hidden state. It can reach significant computational and memory reducing with greater performance loss.

### 2.2 ONE SHOT PRUNING AND TRAINING-BASED PRUNING

Traditional pruning requires re-training after pruning to recover the performance, which is challeng-ing to Large Language Models due to its scale. One-shot pruning can prune LLMs in a single step without need for re-training, reducing the time and computational cost. For example, SparseGPT formalizes the problem of pruning LLMs by solving a local layer-wise reconstruction problem and

prunes the weight matrices based on the weight and inverse Hessian of the loss. Wanda prunes the weight matrices based on the product of weight magnitudes and norm of activation. SliceGPT (Ashkboos et al., 2024a) utilizes singular value decomposition to prune small singular vectors of activation and thus decrease the dimension of hidden state.

Training-based pruning includes mask update and weight update. ADMM-pruner (Boža) and FISTAPruner (Zhao et al., 2024) convert the reconstruction error of a pruned model to a convex problem and use classic convex optimization algorithm to solve the problem. They both update the weight of the dense model, which may lose knowledge of the dense model. These methods use a small size of calibration data, which is similar to one shot pruning. On the other hand, AST (Huang et al., 2024) proposes a pruning framework to retrain pruned models efficiently, while using more data than other methods. Our work is different from these methods. We do not change the weights of LLM to preserve the knowledge of the dense model and we use the same scale of data as one shot pruning.

### 2.3 OUTLIERS IN LLM

Recent studies have found that LLM has a significant phenomenon of outliers (Puccetti et al., 2022; Kovaleva et al., 2021; Timkey & Van Schijndel, 2021), whose magnitude is much more larger than others'. The outliers occur in both weight and activation. Several works on LLM quantization (Ashkboos et al., 2024b; Liu et al., 2024) have developed to efficiently quantize LLMs with little performance loss. In the field of LLM pruning, OWL (Yin et al., 2023) addresses the emergent outliers in LLMs and provides a new technique that leverages the distribution of outliers to guide layer-wise sparsity assignment of LLM pruning. Our work further explore the application of outlier distribution of LLMs. We develop a method to produce more outliers and larger variance in the weight and activation of LLM and therefore improve the performance.

## 3 METHOD

In this section, we will introduce our method. First, we present our method motivated by outliers in LLMs. Next, we describe our method to combine orthonormal matrix with network pruning. Finally, we describe how to train the orthonormal matrix.

### 3.1 MOTIVATION

In network quantization, researchers have focused on the outliers of activation. Removal of outliers improves the performance of quantized networks (Ashkboos et al., 2024b; Liu et al., 2024). In the case of pruning, where weights are eliminated, we focus on weight outliers. It is intuitive that preserving the outliers means small weights in matrix are eliminated, leading to small changes to the weights matrix, and therefore can keep the performance of the network. Based on this motivation, pruning methods can obtain better results by defining better pruning metric to distinguish outliers in LLMs. However, for a specific weight matrix, there must be a best sparsity mask that can achieve the best result of pruning. We want to show that changing the distribution of weight matrix can get a better result.

Consider a linear layer $\boldsymbol{W}$ and input $\boldsymbol{X}$. The result of the linear layer is $\boldsymbol{XW}$. In previous pruning methods, sparsity mask $\boldsymbol{M}$ is obtained based on $\boldsymbol{W}$ and $\boldsymbol{X}$, and the result of pruned linear layer is $\boldsymbol{X}(\boldsymbol{M} \odot \boldsymbol{W})$. If we introduce a matrix $\boldsymbol{A}$ with full rank and apply it to $\boldsymbol{X}$ and $\boldsymbol{W}$, $\boldsymbol{X}' = \boldsymbol{XA}^{-1}$, $\boldsymbol{W}' = \boldsymbol{AW}$, the result of the dense layer is not changed. However, we can get a different pruning result $\boldsymbol{X}'(\boldsymbol{M}' \odot \boldsymbol{W}')$. Consider a simple example:

$$\boldsymbol{X} = \begin{bmatrix} 2 & 1 \end{bmatrix}, \boldsymbol{W} = \begin{bmatrix} \dfrac{1}{\sqrt{2}} & \dfrac{1}{\sqrt{2}} \\ -\dfrac{1}{\sqrt{2}} & \dfrac{1}{\sqrt{2}} \end{bmatrix}$$

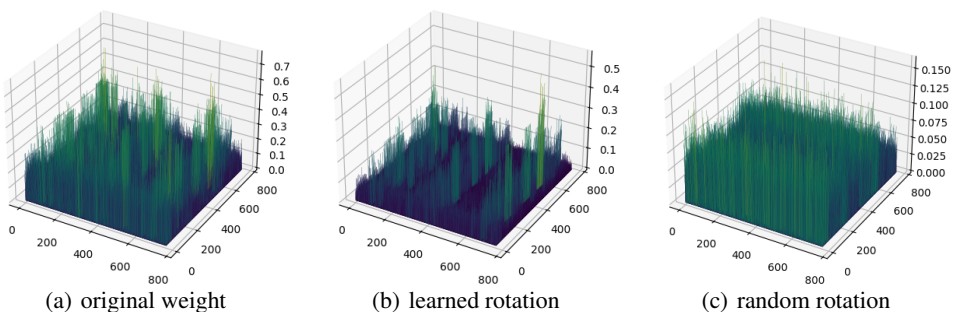

|                     | (a) original weight | (b) learned rotation | (c) random rotation |

Figure 2: Distribution of weights in original space, learned rotated space and random rotated space.

Given the sparsity ratio 50%, it's obvious that pruning on $W$ will always have a wrong result. However, if we apply a transformation,

$$A = \begin{bmatrix} \dfrac{1}{\sqrt{2}} & \dfrac{1}{\sqrt{2}} \\ \dfrac{1}{\sqrt{2}} & -\dfrac{1}{\sqrt{2}} \end{bmatrix} = A^{-T}, \; X' = XA^{-1} = \begin{bmatrix} \dfrac{3}{\sqrt{2}} & \dfrac{1}{\sqrt{2}} \end{bmatrix}, \; W' = AW = \begin{bmatrix} 0 & 1 \\ 1 & 0 \end{bmatrix}$$

pruning on $W'$ does not change the matrix, and thus does not change the result of matrix multiplication.

This simple motivated example implies that pruning on a weight matrix that have more zeros is better. If we can find an $A$ with full rank that minimize $\|AW\|_0$, we can get the best result of pruning. However this problem is hard to solved. Moreover, the optimal $L^0$ norm may not satisfy the required sparsity ratio. This problem can be approximately converted to minimize $\|AW\|_1$ (Candès et al., 2006), which can produce small elements in the weights. But eliminating small elements of $W'$ can harm the performance of the pruned model. The reason is that the distribution of activation is also important to the result of pruned model. In LLMs, the activations are not uniformly distributed, but also have emergent outliers (Ashkboos et al., 2024b).

Figure 2 shows the distribution of the weight matrices in a OPT-125M. We find that the distribution shows the occurrence of outliers. Random rotation will reduce the outliers and harm the performance of pruning (see Table 1). But with specially learned transform, the distribution shows more outliers and produce larger variance.

| Model | OPT-125M | OPT-1.3B | OPT-2.7B | OPT-6.7B |
|---|---|---|---|---|
| original space | 37.41 | 19.01 | 14.60 | 12.38 |
| random rotated space | 12783 | 17786 | 48.40 | 24.61 |

Table 1: WikiText2 perplexity ($\downarrow$) of runing Wanda in original space and rotated space.

## 3.2 ROTATION

Based on the analysis above, we introduce our method to combine orthonormal matrix with network pruning.

We first restrict the transform matrix $A$ to orthonormal matrix (denote as $Q$ below). This can introduce two advantages. Firstly, it's easy to calculate the reversal of $Q$ ($Q^{-1} = Q^T$). Secondly, purely applying $Q$ introduces additional parameters. Based on the computational invariant in Transformer (Vaswani, 2017) from SliceGPT (Ashkboos et al., 2024a), the orthonormal matrix can be fused into linear layers with a RMSNorm connected. The first step of our method is to convert LayerNorm (Ba, 2016) to RMSNorm (Zhang & Sennrich, 2019) (if the norm is LayerNorm) and fuse the scaling coefficient to weight matrices.

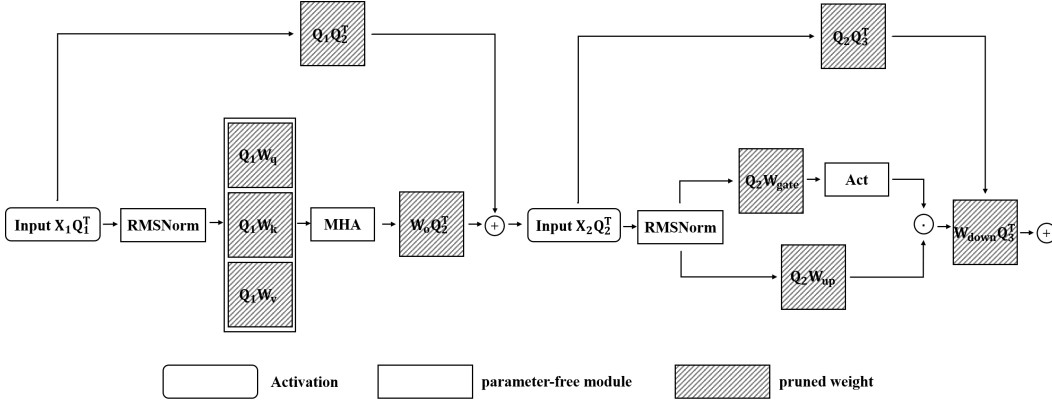

Figure 3: LLaMA block in rotated space. Input matrices $\boldsymbol{W}_{in}$ are pre-multiplied by $\boldsymbol{Q}^T$ and out matrices $\boldsymbol{W}_{out}$ are post-multiplied by $\boldsymbol{Q}$. Layers filled with dotted line are pruned.

In each block of LLM decoder, we introduce the orthonormal matrix $\boldsymbol{Q}$s on the linear layers, as illustrated in Figure 3. Take LLaMA family as an example. There are 7 linear layers in every decoder layer. In each block (self attention or MLP), the weight matrices can be divided into two groups: input matrix $\boldsymbol{W}_{in}$ and output matrix $\boldsymbol{W}_{out}$. Linear layers in the group share the same $\boldsymbol{Q}$ and layers with RMSNorm connected have orthonormal $\boldsymbol{Q}$s. The input matrices are pre-multiplied by $\boldsymbol{Q}^T$ and the out matrices are post-multiplied by $\boldsymbol{Q}$. Specifically, the embedding layer is post-multiplied by $\boldsymbol{Q}$ and the head projection is pre-multiplied by $\boldsymbol{Q}^T$. The total number of $\boldsymbol{Q}$s is two times the number of layers plus one. Once we apply the orthonormal transform to the weight matrix, the pruning method is conducted on them.

Compared with original network, we add an extra matrix multiplication on the residuals, which can increase the size and computation. If the rotation matrices share the same parameters, for example in Figure 3, $\boldsymbol{Q}_1 = \boldsymbol{Q}_2$, the residual is multiplied by an identity matrix, which is the same as the original network. The more rotation matrices share parameters, the size and computation of the pruned network is smaller but the performance is worst. Formally, if there are $l$ layers in the decoder layer and every $k$ $\boldsymbol{Q}$s share parameters, there will be $n = \lfloor 2l/k \rfloor + 1$ individual $\boldsymbol{Q}$s in total. The number of extra parameter is $(n-1) \cdot d^2$ and extra computation is $2(n-1) \cdot bsd^2$, where $b, s, d$ represent the batch size, sequence length and dimension of hidden state. The ablation experiment on shared rotation shows that although this will get a worst result, but can still outperform other method and also save memory and computational cost.

### 3.3 OPTIMIZING ROTATION MATRIX

To train the orthonormal matrix, we define the optimization objective as the performance of the pruned network. The optimization objective comprise the auto-regression training loss of the pruned network. We also add a distillation loss that measures the difference between dense model and sparse model. We consider three types of distillation loss: $L^2$-distance, cosine distance and Jensen–Shannon divergence. The loss is finally defined as:

$$\arg\min \mathcal{L}(\boldsymbol{Q}_i; \boldsymbol{W}_i, \boldsymbol{M}_i, \boldsymbol{X}) = \mathcal{L}_{\text{AR}} + \alpha \mathcal{L}_{\text{distill}}$$

$$\mathcal{L}_{\text{distill}}(Y, \hat{Y}) = \begin{cases} \mathcal{L}_{L^2}(Y, \hat{Y}) = \|Y - \hat{Y}\|_2 \\ \mathcal{L}_{\cos}(Y, \hat{Y}) = 1 - \dfrac{Y \cdot \hat{Y}}{\|Y\|_2 \|\hat{Y}\|_2} \\ \mathcal{L}_{\text{JS}}(Y, \hat{Y}) = D_{\text{KL}}(\theta_Y, \theta_{\hat{Y}}) + D_{\text{KL}}(\theta_{\hat{Y}}, \theta_Y) \end{cases}$$

The pretrained weight $\boldsymbol{W}$ is fixed and we only train the $\boldsymbol{Q}$s. To optimize the orthonormal matrix, we use Cayley SGD method (Li et al., 2020), which can optimize on the Stiefel manifold efficiently.

---

**Algorithm 1** RotPruner

**Inputs:** original model weights $\{W_i\}$ and input $X$, sparsity ratio s, one-shot pruning method $\mathcal{M}$ or fixed masks $\{M_i\}$, epochs
**initialize** $\{Q_i\}$
**for** epoch in epochs **do**
    apply $\{Q_i\}$ on $\{W_i\}$
    **if** given one-shot pruning method $\mathcal{M}$ **then**
        $M_i = \mathcal{M}(W_i')$
    **end if**
    optimize $\{Q_i\}$ to minimize $\mathcal{L}(Q_i; W_i, M_i, X)$ using cayley SGD
**end for**

---

Specifically, in each iteration, the rotation matrices are updated by

$$Q_{k+1} = (I - \frac{\alpha}{2}Y)^{-1}(I + \frac{\alpha}{2}Y)Q_k$$

$Y$ is a skew-symmetric matrix and is chosen to $Y = \hat{Y} - \hat{Y}^T$, where $\hat{Y} = GQ^T - \frac{1}{2}QQ^TGQ^T, G = \nabla f(Q)$. Moreover, the update matrix can be computed via fixed-point iteration to prevent matrix reversing. This optimizer keeps the $Q$s' orthonormality with approximately 2 times of the standard SGD.

When training the model with sparsity matrix $M$, there exists a problem that the gradient cannot be passed through the mask. Previous works use straight-through estimator (STE (Bengio et al., 2013)) to allow gradient to pass through mask by ignoring the mask in the backward pass. The backward of STE can be written as

$$W_{t+1} = W_t - \gamma_t(g(\tilde{W}_t))$$

Sparse-refined straight-through (SR-STE (Zhou et al., 2021)) estimator introduced a sparse-refined regularization term to the gradient, which can prevent mask oscillation. The backward of SR-STE can be written as

$$W_{t+1} = W_t - \gamma_t(g(\tilde{W}_t) + \lambda_W(\bar{M} \odot W_t))$$

We adopt SR-STE to train the orthonormal matrix.

### 3.4 PROCEDURE

Algorithm 1 present the procedure of our method. Our method includes a basic pruning method without weight update and training of orthonormal matrix. For unstructured and semi-structured pruning, we choose two activation-free methods (magnitude based (Han et al., 2015) and Wanda) as the basic pruning method, since they need at most one forward pass and are efficient in time and memory. Activation-based methods need a backward pass and are time consuming. For structured pruning, we fixed the mask to the bottom rows or right columns.

In every training epoch, we apply the orthonormal matrix to the weight matrices and get sparsity masks on every layers based on the basic pruning method. Then we apply the masks and train the orthonormal matrices using STE with sparse-refined regularization term. After several iterations, update the masks based on new orthonormal matrix. For unstructured and semi-structured pruning, the orthonormal matrices are initialized to identity matrix; for structured pruning, they are initialized to the matrices that computed by SliceGPT via Principal Components Analysis.

## 4 EXPERIMENTS

**Models and evaluation** We evaluate our method on widely adopted LLMs: OPT-125M/1.3B/2.7B/6.7B, LLaMA-2-7B, LLaMA-3-8B. We evaluate our method on perplexity and zero-shot task. Perplexity is measured on test set of WikiText-2 (Merity et al., 2016). We use LM Evaluation Harness (Gao et al., 2021) to evaluate zero-shot accuracy on seven benchmarks: Wino-Grande (Sakaguchi et al., 2021), Piqa (Bisk et al., 2020), RTE (Wang, 2018), ARC Easy (Clark et al., 2018), ARC Challenge (Clark et al., 2018), WNLI (Wang, 2018) and QNLI (Wang, 2018).

WinoGrande, Piqa, ARC-easy and ARC-challenge benchmark the ability for knowledge question answering and RTE, QNLI and WNLI benchmark the ability for natural language inference.

**Setup**  We use WikiText2 as the calibration set. We sample 128 data with sequence length 2048. We evaluate three types of pruning: unstructured, 2:4 semi-structured and structured. We train for 5 epochs for each model and set the initial learning rate to 1e-2. The coefficient of SR-STE is 2e-5. We are able to prune an 8B model in an L40S GPU with 48GB memory in 1.5 hour.

**Baseline**  For unstructured and 2:4 semi-structured pruning, we choose SparseGPT and Wanda as baseline. For structured pruning, we compare against SliceGPT.

## 4.1 MAIN RESULTS

### 4.1.1 PERPLEXITY RESULTS

In Table 2, we show the WikiText2 perplexity result for different pruning of various models. We achieved 50% unstructured, 2:4 semi-structured or 30% sturctured sparsity by pruning linear layers in the LLMs, except for embeddings and head. These results show that RotPruner surpasses existing methods on perplexity. We also find that the pruned OPT-6.7B can outperform the dense model.

We also run experiments OPT-1.3B with different sparsity ratio to further analyse the performance of RotPruner. The results are shown in figure Figure 4. We find that our method can surpass the dense model when sparsity ratio is smaller than 50%.

| Method | Sparsity | OPT | | | | LLaMA-2 | LLaMA-3 |
|---|---|---|---|---|---|---|---|
| | | 125M | 1.3B | 2.7B | 6.7B | 7B | 8B |
| Dense | 0% | 27.65 | 14.62 | 12.47 | 10.85 | 5.47 | 6.13 |
| SparseGPT | 50% unstructured | 34.12 | 17.48 | 13.43 | 11.61 | 6.46 | **8.29** |
| Wanda | 50% unstructured | 37.41 | 19.01 | 14.60 | 12.38 | 6.72 | 9.40 |
| RotPruner | 50% unstructured | **30.45** | **14.95** | **13.05** | **10.41** | **6.42** | 8.50 |
| SparseGPT | 2:4 semi-structured | 60.02 | 23.83 | 17.20 | 14.13 | 10.37 | 14.65 |
| Wanda | 2:4 semi-structured | 80.32 | 28.25 | 21.25 | 15.90 | 11.34 | 21.21 |
| RotPruner | 2:4 semi-structured | **43.09** | **17.34** | **16.31** | **13.01** | **9.20** | **11.65** |
| SliceGPT | 30% structured | 42.32 | 20.26 | 16.30 | 12.80 | 8.62 | 17.08 |
| RotPruner | 30% structured | **32.56** | **16.19** | **14.06** | **12.09** | **8.34** | **15.26** |

Table 2: WikiText2 perplexity (↓) performance comparison for different pruning methods on LLMs

### 4.1.2 ZERO-SHOT TASKS

In Table 3, we present the results of zero-shot tasks of different pruning method on OPT-6.7B, LLaMA-2-7B and LLaMA-3-8B. RotPruner surpasses other methods on OPT-6.7B and LLaMA-3-8B on the average accuracy of zero-shot tasks.

## 4.2 INFERENCE SPEED

We evaluate the inference speed of our pruned models on RTX4090. As discussed in subsection 3.2, since we add extra parameters and matrix multiplications on the residuals, we do this experiment to test how these parameters affect the inference speed. We only test on semi-structured pruning, because unstructured pruning has no benefit to the inference speed and we hold the same setting of structured pruning as SliceGPT.

We use Torch's to_sparse_semi_structured() to accelerate the 2:4 structured sparse models and Torch's Timer to benchmark the inference time. We test a LLaMA layer on eight situations: dense layer and sparse layer with residual rotation on attention or MLP. The result in show in Table 4. Tick means adding the residual rotation and cross means not. Specially, dense model without any residual rotations is the original dense model, sparse model without any residual rotations is the sparse

| Model & sparsity | Method | WinoGrand | Piqa | RTE | ARC-e | ARC-c | WNLI | QNLI | Mean |
|---|---|---|---|---|---|---|---|---|---|
| | Dense | 65.51 | 76.27 | 50.50 | 65.66 | 30.46 | 46.48 | 50.87 | 55.81 |
| OPT-6.7B 50% | SparseGPT | **62.95** | 73.18 | 46.43 | 62.96 | **29.09** | 43.66 | 49.46 | 53.05 |
| | Wanda | 61.40 | 72.47 | 44.49 | 62.26 | 27.56 | 43.66 | 49.46 | 51.97 |
| | RotPruner | 62.72 | **74.01** | 45.42 | **64.20** | 28.84 | **45.07** | 49.46 | **53.37** |
| | Dense | 69.22 | 78.07 | 62.82 | 76.30 | 43.43 | 45.07 | 49.97 | 60.70 |
| LLaMA2-7B 50% | SparseGPT | **69.22** | 76.22 | 53.07 | **72.94** | 39.51 | 42.25 | 49.46 | 57.52 |
| | Wanda | 67.88 | **76.77** | 53.43 | 72.85 | **39.84** | 43.66 | 52.21 | **58.09** |
| | RotPruner | 65.27 | 73.99 | **54.15** | 68.85 | 37.03 | **43.66** | 55.45 | 56.91 |
| | Dense | 72.61 | 79.65 | 69.68 | 80.09 | 50.51 | 49.30 | 49.95 | 64.54 |
| LLaMA3-8B 50% | SparseGPT | **70.96** | 74.76 | 55.66 | **72.43** | 39.25 | **43.66** | 49.46 | 58.02 |
| | Wanda | 69.92 | 74.37 | 58.84 | 71.97 | **39.85** | 42.25 | 50.19 | 58.20 |
| | RotPruner | 67.40 | **75.35** | **63.53** | 71.76 | 38.99 | 40.85 | **50.41** | **58.33** |

Table 3: Zero-shot tasks accuracy (↑)

| attention residual rotation | MLP residual rotation | Dense(ms) | Sparse(ms) |
|---|---|---|---|
| ✗ | ✗ | 6.210 | 4.593 (1.352x) |
| ✓ | ✗ | 6.638 | 4.619 (1.344x) |
| ✗ | ✓ | 6.637 | 4.747 (1.308x) |
| ✓ | ✓ | 7.045 | 4.837 (1.284x) |

Table 4: Inference speed of sparse models

model obtained by traditional pruning method. Number in the round brackets means the speedup ratio compared with the dense model.

We find that the extra multiplications slow down the inference time but the gap is not significant especially with only attention residual rotation (1.006 times the time of no residual rotation). This is because other operations in attetion and MLP cost the majority of time and lower the affect of the residual rotation. Therefore, the residual rotation has little affect to the inference speed especially when we decrease the number of $Q$s and add the residual rotation to the attention. If we use half a quarter of $Q$s, which means every four $Q$s share the same parameter, the inference time will be (1.003 times the time of no residual rotation). It's close to the sparse model using traditional pruning methods. Using less orthogonormal matrices will get closer to the traditional pruning methods. Actually, this operation not only has little affect to the performance of the sparse model, but also accelerate the convergence speed. We will provide experiment in the ablation experments.

## 4.3 ABLATION EXPERIMENTS

We conduct ablation experiments to evaluate the performance of our method with different training losses, calibration set size, number of $Q$s and basic pruning methods. The experiments are all conducted on a small model OPT-1.3B for short training time.

| AR | $L^2$ | cosine distance | JS-divergence | result |
|---|---|---|---|---|
| ✓ | | | | 15.00 |
| | ✓ | | | 20.01 |
| | | ✓ | | 18.46 |
| | | | ✓ | 18.80 |
| ✓ | ✓ | | | 14.85 |
| ✓ | | ✓ | | **14.33** |
| ✓ | | | ✓ | 15.03 |

Table 5: Ablation on training loss. Tick means using this loss.

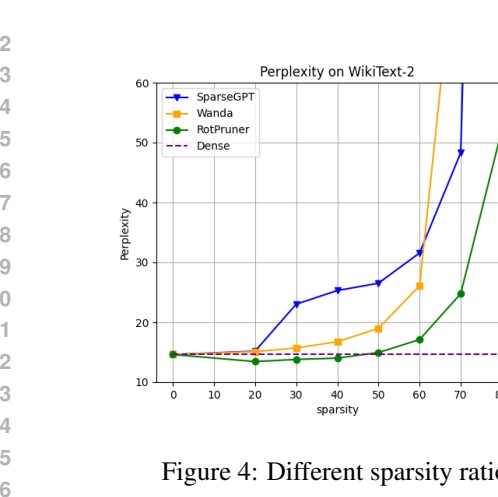 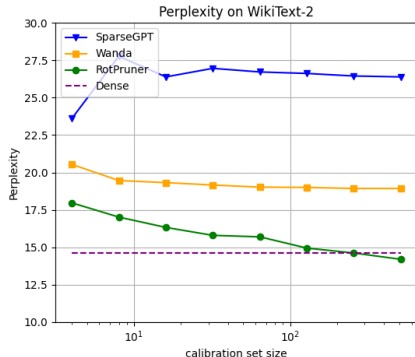

Figure 4: Different sparsity ratio      Figure 5: Different calibration set size

| w/o STE | w STE | w SR-STE($\lambda_W = 2e - 4$) |
|---------|-------|-------------------------------|
| 18.25 | 14.89 | 14.86 |

Table 6: Ablation on Straight Through Estimator.

**Loss**   We study how the training loss influence the results. We have introduced two parts of the loss: auto-regression training loss and distillation loss, and distillation loss can be categorized to $L^2$ distance, cosine distance and JS-divergence. The results are shown in Table 5. From the result, we observe the combination of auto-regression loss and cosine distance perform best in the settings. Therefore, for other experiments, we use this loss by default.

**Straight Through Estimator**   We do ablation experiments on STE(Straight through estimator). We test the performance without STE, with STE and with SR-STE. The results is shown in Table 6. The performance of SR-STE is best among these three condition.

**Optimization Method**   Since we need to optimize on the steifold manifold (which means keeping the matrix's orthogonormality), we use efficient steifold manifold optimization method. We test Cayley Adam. The result is shown in Table 7. Cayley Adam perform worse than Cayley SGD, and it takes more time and memory. Therefore we use Cayley SGD as the optimization method.

**Calibration set size**   We study the calibration set size. Results are shown in Figure 5. We find that compared with other methods, RotPruner is more sensitive to calibration set size. Since using more calibration data increases the memory and computational cost and 128 samples can perform well, we use 128 samples for other experiments.

**Number of $Q$s**   We study how the number of $Q$s affect the result. Theoretically, the more the $Q$s, the better the performance but worse the memory and computation. The maximum number of $Q$s is two times the decoder layers plus one. In OPT-1.3B, which has 24 decoder layer, the maximum number of $Q$s is 49. The result is shown in Table 8. We find that using less $Q$s can get a worst result, but can still outperform other methods. For LLMs with larger scale, we suggest to use less number of $Q$s to save memory and computation cost.

| Cayley SGD | Cayley Adam |
|------------|-------------|
| 14.86 | 15.65 |

Table 7: Ablation on Optimization method.

| number of $Q$s | 49 | 25 | 13 | 7 | 4 |
|---|---|---|---|---|---|
| perplexity | 14.95 | 15.05 | 15.21 | 15.43 | 15.58 |

Table 8: Ablation on number of $Q$s

| method | OPT-1.3B | OPT-2.7B | LLaMA-2-7B |
|---|---|---|---|
| magnitude | 1712 | 265.14 | 19.94 |
| magnitude+RotPruner | **17.15** | **21.08** | **18.84** |
| SparseGPT w/o WR | 23.01 | 23.27 | 7.89 |
| SparseGPT w/o WR+RotPruner | **15.20** | **14.26** | **7.32** |
| Wanda | 19.01 | 14.60 | 6.72 |
| Wanda+RotPruner | **14.86** | **13.05** | **6.42** |

Table 9: Ablation on different basic pruning method. WR stands for weight reconstruction.

**Basic pruning method**    For pruning method, we choose magnitude, Wanda and SparseGPT. Magnitude and Wanda are fast, while SparseGPT takes a long time for pruning and it consists of pruning and weight construction. To shorten the training time, we use SparseGPT without weight reconstruction so that the sparse mask can be fixed. We do experiment on OPT-1.3B, OPT-2.7B and LLaMA-2-7B on the setting of 50% unstructured pruning. The experimental results Table 9 show that our method can improve the performance of one-shot pruning methods.

## 5    CONCLUSION

In this work, we introduce RotPruner, a novel training-based pruning framework for large language models. Unlike traditional pruning methods that operate directly in the original weight and input spaces, RotPruner employs a rotation that transforms the weight matrices and activations into an optimal space for pruning. By doing this, RotPruner enables more effective pruning while minimizing the performance degradation typically associated with model compression. Our approach is compatible with existing pruning methods, allowing for unstructured, semi-structured, and structured pruning.

We evaluate RotPruner on several large language models, including OPT, LLaMA-2, and LLaMA-3, achieving state-of-the-art results in both language modeling and various zero-shot tasks. These evaluations demonstrate RotPruner's ability to enhance pruning performance. The results also highlight the significance of choosing an appropriate transformation space when applying pruning techniques. We hope this work can enhance the understanding of pruning in LLMs and the idea of rotated weight space can help improving the efficiency of neural networks.

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

## A APPENDIX

### A.1 EXPERIMENT ON MORE LLMS

We also add other LLMs: Phi-3-3.8B (Abdin et al., 2024), Mistral-7B (Jiang et al., 2023) and Qwen2-7B (Yang et al., 2024) for comparison. The results are shown in Table 10. Our method perform better in most of the models and pruning settings.

| Method | Sparsity | Phi-3-3.8B | Mistral-7B | Qwen2-7B |
|---|---|---|---|---|
| Dense | 0% | 6.01 | 5.25 | 7.15 |
| SparseGPT | 50% unstructured | 8.56 | 5.99 | **7.92** |
| Wanda | 50% unstructured | 9.37 | 6.28 | 8.40 |
| RotPruner | 50% unstructured | **8.45** | **5.93** | 8.12 |
| SparseGPT | 2:4 semi-structured | **11.99** | 8.16 | 9.30 |
| Wanda | 2:4 semi-structured | 20.00 | 10.70 | 12.19 |
| RotPruner | 2:4 semi-structured | 12.53 | **7.31** | **9.25** |
| SliceGPT | 30% structured | 10.65 | 8.94 | 10.73 |
| RotPruner | 30% structured | **10.16** | **7.42** | **9.64** |

Table 10: WikiText2 perplexity ($\downarrow$) performance comparison for different pruning methods on LLMs

