# OpenReview forum: "RotPruner: Large Language Model Pruning in Rotated Space"
_ICLR.cc/2025/Conference — Submitted to ICLR 2025_

### Official Review · Reviewer_RRj3 · 2024-10-31

**Soundness:** 3
**Presentation:** 3
**Contribution:** 3
**Rating:** 6
**Confidence:** 3

**Summary:**

This paper presents RotPruner, a novel framework designed to improve the pruning of large language models (LLMs) by rotating weight and activation spaces to optimize pruning performance. Unlike traditional pruning techniques that operate directly in the original parameter space, RotPruner transforms this space into a rotated version that enhances pruning effectiveness. Tested on models such as OPT, LLaMA-2, and LLaMA-3, RotPruner achieves superior results over state-of-the-art pruning methods across multiple language benchmarks.

**Strengths:**

1. The paper introduces a simple yet effective method, RotPruner, which enhances model pruning by rotating the weight and activation spaces in linear layers, significantly improving pruning performance.
2. Extensive experiments validate the general effectiveness of RotPruner, with tests conducted on three different LLM series and across eight benchmarks.
3. The theoretical derivations are well-structured and accessible, making the methodology easy to understand.
4. The authors provide complete experimental code, ensuring reproducibility and facilitating future research in this area.

**Weaknesses:**

1. The authors do not provide experiments to assess whether RotPruner introduces additional overhead in training or inference.

2. Although the authors claim that RotPruner is orthogonal to other pruning methods, the paper lacks detailed discussion or evidence to substantiate this claim.

**Questions:**

1. How would different methods of learning the rotation matrices affect the results?

---

> ### Author Response · Authors · 2024-11-23
> **Official Comment by Authors**
>
> Thank you for your insightful feedback. We have integrated your suggestions into the revision. Below is a detailed response to each of your questions.
>
> # Response to weakness 1
>
> We evaluate the inference speed of our pruned models on RTX4090. As discussed in subsection 3.2, since we add extra parameters and matrix multiplications on the residuals, we do this experiment to test how these parameters affect the inference speed. We only test on semi-structured pruning, because unstructured pruning has no benefit to the inference speed and we hold the same setting of structured pruning as SliceGPT.
>
> We use Torch's `to_sparse_semi_structured()` to accelerate the 2:4 structured sparse models and Torch's `Timer`to benchmark the inference time. We test a LLaMA layer with Flash Attention implemented on eight situations: dense layer and sparse layer with residual rotation on attention or MLP. The result in show below. Tick means adding the residual rotation and cross means not. Specially, dense model without any residual rotations is the original dense model, sparse model without any residual rotations is the sparse model obtained by traditional pruning method. Number in the round brackets means the speedup ratio compared with the dense model.
>
> | attention residual rotation | MLP residual rotation | Dense(ms) |     Sparse(ms)     |
> | :-------------------------: | :-------------------: | :---: | :------------: |
> |          &#10007;           |       &#10007;        | 6.210 | 4.593 (1.352x) |
> |          &#10004;           |       &#10007;        | 6.638 | 4.619 (1.344x) |
> |          &#10007;           |       &#10004;        | 6.637 | 4.747 (1.308x) |
> |          &#10004;           |       &#10004;        | 7.045 | 4.837 (1.284x) |
>
> We find that the extra multiplications slow down the inference time but the gap is not significant especially with only attention residual rotation (1.006 times the time of no residual rotation). This is because other operations in attetion and MLP cost the majority of time and lower the affect of the residual rotation. Therefore, the residual rotation has little affect to the inference speed especially when we decrease the number of $Q$s and add the residual rotation to the attention. If we use half a quarter of $Q$s, which means every four $Q$s share the same parameter, the inference time will be (1.003 times the time of no residual rotation). It's close to the sparse model using traditional pruning methods. Actually, this operation not only has little affect to the performance of the sparse model, but also accelerate the convergence speed. We provide experiments in the ablation experments.
>
> # Response to weakness 2
>
> As we discussed in Algorithm 1, this method can use any pruning method to get the pruning performance of the rotated space. We choose Wanda as the pruning method in most of the experiments, since Wanda is fast and has good performance. If using other pruning methods, it takes more time to get the result of the sparse model and makes the training of RotPruner longer.
>
>
>
> # Response to question 1
>
> We add additional ablation experiments to study how different methods of training the rotation matrices affect the results. Since we need to optimize on the steifold manifold (i.e. keeping the matrix's orthogonormality), we use efficient steifold manifold optimization method. We test Cayley Adam on OPT-1.3B. The result is shown below. Cayley Adam performs worse than Cayley SGD, and it takes more time and memory. Therefore we use Cayley SGD as the optimization method.
>
> | Cayley SGD | Cayley Adam |
> | ---------- | ----------- |
> | 14.95      | 15.65       |
>
> We also do ablation experiments on STE(Straight through estimator). We test the performance without STE, with STE and with SR-STE. The results is shown below. The performance of SR-STE is best among these three condition.
>
> | w/o STE | w STE | w SR-STE($\lambda_W=2e-4$) |
> | ------- | ----- | -------------------------- |
> | 18.25   | 14.89 | 14.95                      |

---

> > ### Comment · Reviewer_RRj3 · 2024-11-23
> >
> > Thanks for the author's reply. Do you have any experimental results on pruning methods other than Wanda?

---

> > > ### Author Response · Authors · 2024-11-24
> > >
> > > We choose three one-shot pruning methods: magnitude, Wanda and SparseGPT. Magnitude and Wanda are fast, while SparseGPT takes a long time for pruning and it consists of pruning and weight construction. To shorten the training time, we use SparseGPT without weight reconstruction so that the sparse mask can be fixed. We do experiment on OPT-1.3B, OPT-2.7B and LLaMA-2-7B on the setting of 50% unstructured pruning. The experimental results show that our method can improve the performance of one-shot pruning methods.
> > >
> > > |                                     | OPT-1.3B  | OPT-2.7B  | LLaMA-2-7B |
> > > | ----------------------------------- | --------- | --------- | ---------- |
> > > | Magnitude                           | 1712      | 265.14    | 19.94      |
> > > | Magnitude+RotPruner                 | **17.15** | **21.08** | **18.84**  |
> > > | SparseGPT w/o weight reconstruction | 23.01     | 23.27     | 7.89       |
> > > | SparseGPT+RotPruner                 | **15.20** | **14.26** | **7.32**   |
> > > | Wanda                               | 19.01     | 14.60     | 6.72       |
> > > | Wanda+RotPruner                     | **14.86** | **13.05** | **6.42**   |

---

### Official Review · Reviewer_16o6 · 2024-11-04

**Soundness:** 3
**Presentation:** 3
**Contribution:** 2
**Rating:** 5
**Confidence:** 4

**Summary:**

The paper systematically proposes that the original weight space of a large model is not necessarily the optimal pruning space, and proposes to find the optimal pruning space by rotating matrix Q. The pruning operation is carried out in the rotated weight space, and to the greatest extent, the performance of the original model is retained under the same sparsity of the pruning operation.

**Strengths:**

1. Innovative thinking on pruning space: The paper systematically proposes that the original weight space of a large model is not necessarily the optimal pruning space, and proposes to find the optimal pruning space by rotating matrix Q. The pruning operation is carried out in the rotated weight space, and to the greatest extent, the performance of the original model is retained under the same sparsity of the pruning operation.
2. Outstanding algorithm performance: The algorithm used in the paper has excellent performance in the experiments listed in the text.
3. concise language and clear logic: the text is presented in a concise and logical manner, and the experimental results are well organized to help readers clearly understand its research contributions.

**Weaknesses:**

1. Limited model and dataset, too few comparison algorithms: the experiment only compares two pruning algorithms, Wanda and SparseGPT, and the model only chooses OPT and LLAMA series, which is not convincing enough
2. Single evaluation index: only PERPLEXITY was used as the performance index of the large model after pruning, more indexes can be introduced to verify the effectiveness of the evaluation algorithm.
3. ablation experiment design cannot verify the effectiveness of the algorithm: ablation experiments evaluate the effect of different losses on the algorithm, but the losses themselves are not the focus of the article's discussion.
4. Missing reasonableness analysis to choose compare the zero-shot learning ability : the experiments assess the performance of the algorithm when comparing the different pruning algorithms in the zero-sample learning ability of the difference, but the text does not mention at all the reasonableness analysis of the choice of the index, and does not mention the algorithm in the pruning of the model after the zero-sample learning ability of the performance of the explanation.
5, the experimental design is insufficient: the experimental part is missing the analysis of the effect of pruning in the rotated weight space and the original weight space, resulting in a lack of rigor in the experimental verification of the effectiveness of the algorithm and a lack of explanatory power.

**Questions:**

I don't have questions.

---

> ### Author Response · Authors · 2024-11-23
> **Official Comment by Authors**
>
> Thank you for your insightful feedback. We have integrated your suggestions into the revision. Below is a detailed response to each of your questions.
>
> # Response to weakness 1
>
> We add other LLMs: Phi-3-3.8B, Mistral-7B and Qwen2-7B to make the results more comprehensive. Our method perform better in most of the models and pruning settings.
>
> | Method    | Sparsity           | Phi-3-3.8B | Mistral-7B | Qwen2-7B |
> | --------- | ------------------ | ---------- | ---------- | -------- |
> | Dense     | 0%                 | 6.01       | 5.25       | 7.15     |
> | SparseGPT | 50% unstructured   | 8.56       | 5.99       | **7.92** |
> | Wanda     | 50% unstructured   | 9.37       | 6.28       | 8.40     |
> | RotPruner | 50% unstructured   | **8.45**   | **5.93**   | 8.12     |
> | SparseGPT | 2:4 semi-strutured | **11.99**  | 8.16       | 9.30     |
> | Wanda     | 2:4 semi-strutured | 20.00      | 10.70      | 12.19    |
> | RotPruner | 2:4 semi-strutured | 12.53      | **7.31**   | **9.25** |
> | SliceGPT  | 30% unstructured   | 10.65      | 8.94       | 10.73    |
> | RotPruner | 30% unstructured   | **10.16**  | **7.42**   | **9.64** |
>
> # Response to weakness 2
>
> The evaluation indexes we choose refer to other works in LLM pruning (SparseGPT, Wanda, SliceGPT, etc). For language modeling benchmark, these works focus on perplexity.
>
> # Response to weakness 3
>
> Loss design is a part of the Algorithm 1 so we do ablation experiments on it. We add additional ablation experiments to show the effectiveness of Algorithm 1.
>
> We add additional ablation experiments to study how different methods of training the rotation matrices affect the results. Since we need to optimize on the steifold manifold (i.e. keeping the matrix's orthogonormality), we use efficient steifold manifold optimization method. We test Cayley Adam on OPT-1.3B. The result is shown below. Cayley Adam performs worse than Cayley SGD, and it takes more time and memory. Therefore we use Cayley SGD as the optimization method.
>
> | Cayley SGD | Cayley Adam |
> | ---------- | ----------- |
> | 14.95      | 15.65       |
>
> We also do ablation experiments on STE(Straight through estimator). We test the performance without STE, with STE and with SR-STE. The results is shown below. The performance of SR-STE is best among these three condition.
>
> | w/o STE | w STE | w SR-STE($\lambda_W=2e-5$) |
> | ------- | ----- | -------------------------- |
> | 18.25   | 14.89 | 14.86                      |
>
> # Response to weakness 4
>
> We choose seven benchmark to evaluate the zero-shot performance. WinoGrande, Piqa, ARC-easy and ARC-challenge benchmark the ability for knowledge question answering and  RTE, QNLI and WNLI benchmark the ability for natural language inference.
>
> # Response to weakness 5
>
> Pruning in the original weight space is to run the baseline methods. You can refer to the baseline method as the result of pruning in the original weight space.

---

### Official Review · Reviewer_rYD4 · 2024-11-04

**Soundness:** 2
**Presentation:** 2
**Contribution:** 2
**Rating:** 5
**Confidence:** 4

**Summary:**

This paper introduces RotPruner, an adaptable pruning framework for LLMs that prunes model parameters in the rotated representation space. RotPruner adds orthonormal matrices to LLMs and converts model hidden states to a rotated spade for parameter pruning, thus preserving more model outliers and keeping the LLM's knowledge learned in pretraining. The method is adaptable to existing LLM pruning methods with structured, semi-structured, and unstructured pruning strategies. Experimental results show that RotPruner outperforms existing pruning baselines.

**Strengths:**

- This paper presents a novel idea of pruning LLM parameters in a rotated space, thus preserving more outlier parameters with learned capabilities.
- The RotPruner method is compatible with existing pruning methods on a wide range of setups (structured, semi-structured, and unstructured)

**Weaknesses:**

- There is no strong motivation for pruning LLM parameters in the rotated space. I feel like the author should have a better discussion of this in the introduction section.
- As RotPruner adds orthonormal matrices to the LLM to facilitate pruning, there should be a model latency and memory consumption overhead compared to traditional pruning methods, but the authors didn't include these comparisons in the paper.
- The authors imply that RotPruner is a post-training pruning method, yet Algorithm 1 shows that the pruned model needs to be further re-trained (approx. 1.5 hours) to recover the model performance. It makes the comparison to SparseGPT and Wanda (few-shot and tuning-free pruning methods) unfair.
- There are a lot of typos and confusion in this paper (details in the following Questions section)

**Questions:**

Questions:
- What is the author's motivation for pruning LLMs in the rotated space? Is there a theoretical analysis that could demonstrate that pruning in the rotated space is better than doing it in the original model parameter space?
- Why does pruning LLM parameters in the rotated space preserve more outliers?
- What is the latency and memory consumption of RotPruner-pruned models? How is it compared to those models pruned with traditional pruning methods?
Typos and confusions:
- Please add citations to SparseGPT, Wanda, and SliceGPT when they first occur in the last paragraph of the introduction section.
- Figure 2 is distracting as the audience of this paper should already know what unstructured, semi-structured, and structured pruning methods are. Since they are not this paper's main contributions, I recommend the author remove this figure.
- line 256: "in 4", and I suspect that it should be "in Figure 4"
- line 208: "see table 1" (and lots of the same typos in the following). Please capitalize the words "Table," "Figure," etc. Consider using "\cref".
- Table 2 and Table 3: I suspect that 50% sparsity corresponds to the unstructured pruning setup and 30% to the structured pruning setup, yet this information is missing from both tables.
- Table 4: Using ticks to represent ablations is super confusing. I don't know if a tick means a module/functionality is removed or preserved.

---

> ### Author Response · Authors · 2024-11-23
> **Official Comment by Authors**
>
> Thank you for your insightful feedback. We have integrated your suggestions into the revision. Below is a detailed response to each of your questions.
>
> # Response to question 1
>
> Our motivation is that pruning in a transformed space has a probability to get a better performance after pruning. We give an example to show this. However, it's hard to get a optimal space theoretically, therefore we propose a method to get this transformation through learning. We choose the transformation as rotation for the reason that it's easy to get the reversal of the matrix and can be fused into LayerNorm.
>
> # Response to question 2
>
> We give an example of OPT-125m to show that rotating the space can generate more outliers. But in LLMs of larger scale, it's hard to observe this phenomenon, and there are small differences between rotation matrices and identity matrix. This means larger LLMs have more outliers and the rotation only slightly change the space, but it can still have a better performance.
>
> # Response to weakness 3
>
> RotPruner takes more time than one-shot pruning method. But the training in Algorithm 1 is different from retraining. Model weights are frozen and only the rotations are trainable. This training is to optimize rotation matrices. Recover finetune takes more time, data and memory. For example, in Wanda, it takes 30000 samples and 12 hours to run the recover finetuning. We use the same scale of data as other post-training methods. Our method can be seen as an efficient way to recover the model performance.
>
> # Response to question 3
>
> We evaluate the inference speed of our pruned models on RTX4090. As discussed in subsection 3.2, since we add extra parameters and matrix multiplications on the residuals, we do this experiment to test how these parameters affect the inference speed. We only test on semi-structured pruning, because unstructured pruning has no benefit to the inference speed and we hold the same setting of structured pruning as SliceGPT.
>
> We use Torch's `to_sparse_semi_structured()` to accelerate the 2:4 structured sparse models and Torch's `Timer`to benchmark the inference time. We test a LLaMA layer with Flash Attention implemented on eight situations: dense layer and sparse layer with residual rotation on attention or MLP. The result in show below. Tick means adding the residual rotation and cross means not. Specially, dense model without any residual rotations is the original dense model, sparse model without any residual rotations is the sparse model obtained by traditional pruning method. Number in the round brackets means the speedup ratio compared with the dense model.
>
> | attention residual rotation | MLP residual rotation | Dense |     Sparse     |
> | :-------------------------: | :-------------------: | :---: | :------------: |
> |          &#10007;           |       &#10007;        | 6.210 | 4.593 (1.352x) |
> |          &#10004;           |       &#10007;        | 6.638 | 4.619 (1.344x) |
> |          &#10007;           |       &#10004;        | 6.637 | 4.747 (1.308x) |
> |          &#10004;           |       &#10004;        | 7.045 | 4.837 (1.284x) |
>
> We find that the extra multiplications slow down the inference time but the difference is not significant especially with only attention residual rotation (1.006 times the time of no residual rotation). This is because other operations in attetion and MLP cost the majority of time and lower the affect of the residual rotation. Therefore, the residual rotation has little affect to the inference speed especially when we decrease the number of $Q$s and add the residual rotation to the attention. If we use half a quarter of $Q$s, which means every four $Q$s share the same parameter, the inference time will be (1.003 times the time of no residual rotation). It's close to the sparse model using traditional pruning methods. Actually, this operation not only has little affect to the performance of the sparse model, but also accelerate the convergence speed. We provide experiments in the ablation experments.
>
> # Response to question 4-9
>
> Thanks for pointing out the typos and confusions. We have fixed these in the modified pdf. For Table 2 and Table 3, 50% sparsity corresponds to the unstructured pruning setup and 30% to the structured pruning. For Table 4, ticks means preserving the module.

---

> > ### Comment · Reviewer_rYD4 · 2024-12-03
> >
> > Thanks for the authors' reply. Most of my concerns have been resolved, especially on the inference efficiency side. However, I still disagree that RotPruner has a key difference regarding the LLM training paradigm compared to performance recovery finetuning strategies. A reasonable set of experiments could be setting a perplexity/accuracy target first, and examining the training-time-to-accuracy with different methods (e.g., Wanda, SparseGPT, and RotPruner) to demonstrate that RotPruner is more efficient in training. I would raise my score to 5.

---

### Meta-Review · Area_Chair_xeA3 · 2024-12-20

**Metareview:**

This paper introduces RotPruner, a new framework for pruning large language models (LLMs). The key idea is that instead of pruning model parameters directly in their original weight space, RotPruner applies a rotation (using orthonormal matrices) to transform the representation space (both weights and activations) before pruning. This rotated space allows for more effective identification and removal of less important parameters, preserving more "outliers" and knowledge learned during pre-training. RotPruner is adaptable to various pruning strategies (structured, semi-structured, and unstructured) and has been shown to outperform existing pruning methods on models like OPT and LLaMA models across multiple language benchmarks.

However, reviewers concerned about the lack of overhead analysis for the proposed RotPruner method, including insufficient experimental evaluation (limited models, datasets, comparison algorithms, and evaluation metrics). The  justification for pruning in the rotated space also need more motivation and explanation.

**Additional Comments On Reviewer Discussion:**

After the discussion period, the reviewers agreed the paper is not ready can be improved with addressing the aforementioned weaknesses.

---

### Decision · Program_Chairs · 2025-01-22

Reject